# Deep Subsurface Hypersaline Environment as a Source of Novel Species of Halophilic Sulfur-Oxidizing Bacteria

**DOI:** 10.3390/microorganisms10050995

**Published:** 2022-05-09

**Authors:** Lea Nosalova, Maria Piknova, Katarina Bonova, Peter Pristas

**Affiliations:** 1Department of Microbiology, Faculty of Science, Institute of Biology and Ecology, Pavol Jozef Safarik University in Kosice, 041 54 Kosice, Slovakia; lea.nosalova@student.upjs.sk (L.N.); peter.pristas@upjs.sk (P.P.); 2Faculty of Science, Institute of Geography, Pavol Jozef Safarik University in Kosice, 040 01 Kosice, Slovakia; katarina.bonova@upjs.sk; 3Centre of Biosciences, Institute of Animal Physiology, Slovak Academy of Sciences, 040 01 Kosice, Slovakia

**Keywords:** halophiles, sulfur-oxidizing bacteria, hypersaline environment, deep-subsurface

## Abstract

The sulfur cycle participates significantly in life evolution. Some facultatively autotrophic microorganisms are able to thrive in extreme environments with limited nutrient availability where they specialize in obtaining energy by oxidation of reduced sulfur compounds. In our experiments focused on the characterization of halophilic bacteria from a former salt mine in Solivar (Presov, Slovakia), a high diversity of cultivable bacteria was observed. Based on ARDRA (Amplified Ribosomal DNA Restriction Analysis), at least six groups of strains were identified with four of them showing similarity levels of 16S rRNA gene sequences lower than 98.5% when compared against the GenBank rRNA/ITS database. Heterotrophic sulfur oxidizers represented ~34% of strains and were dominated by *Halomonas* and *Marinobacter* genera. Autotrophic sulfur oxidizers represented ~66% and were dominated by *Guyparkeria* and *Hydrogenovibrio* genera. Overall, our results indicate that the spatially isolated hypersaline deep subsurface habitat in Solivar harbors novel and diverse extremophilic sulfur-oxidizing bacteria.

## 1. Introduction

The study of microbiota inhabiting extreme hypersaline environments has gained increasing interest in recent years due to its incomplete characterization [1]. Hypersaline habitats such as soda lakes, salt mines, or deep-sea salt brines were traditionally considered as environments with low variability of bacteria, colonized mostly by highly adapted *Archaea*. However, it is now becoming clear that these extreme habitats harbor complex microbial communities, including chemolithotrophic sulfur-oxidizing bacteria [2]. High-salt environments are part of global ecosystems contributing significantly to the sulfur cycle [3]. Sulfur-oxidizing bacteria have adapted to life in these hypersaline conditions owing to great energy yield during the oxidation of reduced sulfur compounds [4]. Relatively high cultivable diversity of halophilic sulfur-oxidizing bacteria was discovered in lakes in Mongolia, Russia, Ukraine [5], China [3], and in Mediterranean Sea [6]. In our preliminary experiments, a diverse population of halophilic bacteria was observed in salt brine from the former salt mine in Solivar (Presov, East Slovakia). Surprisingly, 80% of strains were able to grow on thiosulfate medium (data not shown), including genera *Planococcus*, *Chromohalobacter*, *Halovibrio*, *Halomonas*, and *Marinobacter*. To better understand the sulfur-oxidizing microbiota from the deep subsurface hypersaline environment, cultivation experiments were performed using thiosulfate medium as a primary isolation medium.

## 2. Materials and Methods

### 2.1. Sampling Site Characterization and Primary Isolation of Bacteria

Brine samples were collected in sterile tubes on the surface from a salt mine borehole in Solivar (Presov, East Slovakia, GPS: 48°59′0.706″ N, 21°16′59.124″ E) in September 2021 and were transported immediately on ice to the laboratory. The physicochemical parameters, such as pH, conductivity, and temperature, were measured in situ using a portable multimeter HI98129 (Hannah instruments, Woonsocket, RI, USA). Total dissolved solids (TDS) were determined by a gravimetric method under laboratory conditions, and the experiment was performed in triplicate.

In order to isolate sulfur-oxidizing bacteria, one hundred microliters of water samples was inoculated onto thiosulfate agar medium (pH 7.5) containing the following (g/L): NH_4_Cl 0.1, NaHCO_3_ 0.2, K_2_HPO_4_ 0.1, Na_2_S_2_O_3_ 5, agar 15 (Merck, Darmstadt, Germany), supplemented with 5% (*w*/*v*) NaCl, if not stated otherwise. After a seven-day incubation at laboratory temperature (20–25 °C), the number of colony-forming units (CFU) per one milliliter of water sample was determined. All individual colonies were picked and re-streaked onto thiosulfate agar medium several times until pure cultures were obtained. The purity of bacterial strains was checked by examining the colony morphology and Gram staining. All chemicals used for media preparation were purchased from Centralchem (Bratislava, Slovakia).

### 2.2. Cultivation Analyses

For determination of salt dependence and the ability to lead an autotrophic or a heterotrophic lifestyle, strains were cultured either on R2A (Merck, Darmstadt, Germany) or thiosulfate agar media, supplemented with appropriate salt concentration, specifically with 0%, 5%, 10%, or 20% (*w*/*v*) NaCl. After inoculation, the plates were incubated at 25 °C for 2 days on R2A medium or for 7 days on thiosulfate agar medium.

### 2.3. DNA Isolation and Restriction Analysis

Total DNA was extracted from colonies using the modified phenol/chloroform method by Pospiech and Neumann (1995) [7] after two weeks of cultivation on thiosulfate agar medium. The quality of DNA was checked by 1% agarose gel electrophoresis in 1× TAE buffer (Thermo Fisher Scientific, Waltham, MA, USA) and staining with ethidium bromide (0.1 µg/mL). The gels were visualized and imaged using Molecular Imager ChemiDocTM XRS+ (BIO-RAD, Hercules, FL, USA). DNA quantification was performed using the Nanodrop One/OneC Microvolume UV–Vis Spectrophotometer (Thermo Fisher Scientific, Waltham, MA, USA).

Subsequently, the obtained genomic DNA was used as a template for 16S rRNA gene amplification using Taq Core Kit/high yield (Jena Bioscience, Jena, Germany) and universal bacterial primers fD1 (5′-AGAGTTTGATCCTGGCTCAG-3′) and rP2 (5′-ACGGCTACCTTGTTACGACTT-3′) [8]. The PCR was performed in a volume of 50 µL containing: 5 µL 10× reaction buffer, 200 μM dNTP, 1 μM fD1 primer, 1 μM rP2 primer, 2.5 U T4-Taq DNA polymerase (Taq Core Kit, Jena Bioscience, Jena, Germany), sterile deionized water and 50 ng of DNA template. Amplification was performed on Master cycler Pro S (Eppendorf, Hamburg, Germany) under the following conditions: initial denaturation at 94 °C for 5 min, followed by 35 cycles of denaturation at 94 °C for 60 s, annealing at 54 °C for 60 s, and extension at 72 °C for 90 s, and a final extension at 72 °C for 5 min. PCR products were separated in 1% agarose gels and visualized as stated above.

For dereplication of bacterial strains, PCR amplicons were digested separately with restriction endonucleases AluI or HaeIII (Jena Bioscience, Jena, Germany) using the following reagents per 20 µL reaction: 0.5 µg PCR product, 2 µL 10× Universal Buffer, 2–5 U of enzyme, filled with sterile deionized water up to 20 µL. Reaction mixtures were incubated at 37 °C for one hour and separated in 1.5% agarose gel.

### 2.4. 16S rRNA Gene Sequencing and Bioinformatic Analyses

Sequencing of 16S rRNA gene amplicons was performed at SEQme (SEQme Ltd., Dobris, Czech Republic) using the Sanger method. Chromatograms were visually inspected and edited manually where necessary. The sequences were assembled by BioEdit version 7.2.5 [9] and subsequently compared with those available in GenBank rRNA/ITS database using BlastN algorithm (https://www.ncbi.nlm.nih.gov/blast, accessed on 1 May 2022) or against EzTaxon database [10].

Then, 16S rRNA gene sequences obtained in this study were checked for the presence of chimeric sequences using DECIPHER’s Find Chimeras web tool [11] and submitted to the GenBank database under the accession numbers ON028648-ON028654.

Sequence data of the best hits from BlastN searches were downloaded, and all sequences were multiple aligned by ClustalW implemented in the MEGA X software version 10.1.6 [12]. The phylogenetic tree was constructed using the neighbor-joining algorithm with 1000 bootstrap replications.

Based on the results from ARDRA analysis, the strains were divided into several groups, and using the abundance of each group, the Chao1 index [13] was calculated to estimate the richness of cultivable halophilic sulfur-oxidizing bacteria.

## 3. Results and Discussion

The former salt mine Solivar near Presov is located along the eastern margin of the Central Western Carpathians. The first mention of salt springs from this area comes from the 13th century, and the first evidence of mining activity dates to the 16th century [14]. The salt was mined from deposits of the Eastern Slovak Basin representing the north-western depocenter of the Transcarpathian Basin. The basin fill comprises sedimentary rocks belonging to several sedimentary cycles, which are the result of complicated tectonic regime and palaeogeographic changes during the Neogene [15,16]. The basin sediments have a transgressive character and consist of conglomerate, sandstone, and siltstones with a marine microfauna indicating a neritic to shallow bathyal zone and are 17.5–16.4 million years old [17]. The sedimentary cycle ended by an evaporation stage with a deposition of the salt mine formation. The character of deposition and sporadic foraminiferal microfauna indicate evaporate precipitation in saline mud flat and salt-pan environments. Evaporates were deposited as a result of shallowing or precipitation at the water–air interface in the central part of the basin due to its entire isolation [18].

### 3.1. Spring Chemistry

Samples were collected at the point where the flowing brine emerged from a borehole. The sampling site was located in the area of the former salt mine Solivar near Presov, characterized by a salt-bearing formation in a depth from 510 to 600 m below the surface [19].

At the time of sampling procedure, the temperature of salt water was 12.3 °C, conductance > 3999 μS/cm and pH 6.5. Based on gravimetric analysis, the water from a borehole can be considered in a hypersaline environment with an average salinity as high as 311 g/L of total dissolved solids, exceeding the concentration of salt in sea water (35 g/L) almost ten times.

### 3.2. Isolation of Halophilic Bacterial Strains

With the aim of isolating sulfur-oxidizing halophilic bacteria, brine samples were plated on thiosulfate agar medium supplemented with 5% (*w*/*v*) NaCl. After seven days of cultivation, white slightly translucent colonies with the frequency of approximately 212 CFU/mL of brine were observed. The CFU values were comparable to those from our preliminary study using NaCl-supplemented TSA medium for isolation of bacteria (data not shown). The total number of cultivable bacteria detected in this study was one order of magnitude lower than those reported by Maturrano et al. (2006) [20] from a hypersaline environment in the Andes or by Antony et al. (2013) [21] from soda lakes in India. For further analyses, 41 bacterial colonies were randomly selected and obtained in pure cultures by repeated subculturing.

### 3.3. Cultivation Analyses

Salt requirement and tolerance are known to vary according to growth conditions, especially temperature and medium composition [22]. To examine the salt tolerance/dependence as well the capability of autotrophic growth, all strains were grown on media with different salinities (0%, 5%, 10%, and 20% (*w*/*v*) NaCl). All strains were found to be halophiles, as no growth in the absence of salt was observed on both types of cultivation media tested. However, an interesting salt tolerance pattern was observed when comparing both types of media. On defined mineral thiosulfate agar medium, all strains were able to tolerate salt concentrations up to 10%, and some bacterial representatives could even grow on medium with 20% NaCl. Conversely, the complex R2A medium with 10% salt concentration supported the growth of 34% of strains only, and medium with 20% NaCl inhibited the growth of all bacteria tested. From the perspective of autotrophic growth, it can be concluded that 34% of bacterial strains, later identified as members of genera *Halomonas* and *Marinobacter*, were facultatively autotrophic as judged by their ability to grow and obtain energy from heterotrophic R2A medium supplemented with 5% or 10% NaCl. In some cases, production of visible sulfur particles on the surface of colonies was observed after prolonged cultivation (14 days). The formation of elemental sulfur during thiosulfate oxidation has been described for several sulfur oxidizers, e.g., representatives of *Guyparkeria* and *Hydrogenovibrio* [23,24].

### 3.4. Molecular Identification of Halophilic Sulfur-Oxidizing Bacterial Strains

To date, there are only a few papers dealing with the diversity of salt mine microbiota. For example, as reported by Chen et al. (2007) [25], the cultivable microbiota from the Yipinglang salt mine (Yipinglang, China) consisted of more than 20 genera with a predominance of *Proteobacteria*. Another study by Cycil et al. (2020) [26] examined the microbial diversity of the Karak salt mine (Karak, Pakistan), where similar salinity of brines was observed. However, although metagenomic data revealed the presence of several microorganisms participating in the sulfur cycle, more detailed analyses using cultivation approaches are needed to characterize them.

In our study, molecular approaches were used for the identification of halophilic sulfur oxidizers from the Solivar salt brine. In the first step, PCR amplifications of nearly full-length 16S rRNA genes of all bacterial strains were performed. For strain dereplication, the resultant 16S amplicons were subjected to ARDRA analysis (Amplified Ribosomal DNA Restriction Analysis), and based on the observed banding patterns, all 41 strains were divided into six groups (I–VI). The calculated Chao1 richness estimates showed that we covered the whole biodiversity of cultivable halophilic sulfur oxidizers from the Solivar salt brine, since the number of estimated groups was 6.7 with 95% probability and cultivation analyses revealed the presence of six different bacterial groups. To determine the taxonomic affiliation, one representative from each group was selected, and its 16S rRNA gene was sequenced. Comparative analyses of the obtained sequences using the rRNA/ITS database and the EzTaxon database revealed that the isolated bacteria show similarity values ranging from 97.55% to 99.93% to validly described taxa (Table 1).

Overall, the cultivable microbiota identified in this study were composed of bacteria belonging to four different orders within *Gammaproteobacteria*, namely *Alteromonadales*, *Oceanospirillales*, *Thiotrichales*, and *Chromatiales*.

The most prevalent bacterial genus was *Guyparkeria* of the order *Chromatiales*, represented by 18 strains, falling into ARDRA groups I and II. Currently, there are only two validly described *Guyparkeria* species, namely *G. hydrothermalis* and *G. halophila*, both reclassified from the *Halothiobacillus* genus, as their metabolic properties and salt requirements differed [23]. It was reported that both species can oxidize several reduced sulfur compounds, thus influencing the pH of their environment [23,27]. Based on BlastN sequence analysis of the 16S rRNA gene, the closest relative of the ARDRA group I strain 1SP6A2 was identified as *G. hydrothermalis*. However, a low similarity score of 97.63% indicated that this bacterial strain represents a prospective novel *Guyparkeria* species.

The order *Alteromonadales* was represented by only three strains forming ARDRA group III, which were identified as *Marinobacter aquaticus* with a sequence similarity of 99.49%. *Marinobacter spp.* are ubiquitously found in different types of hypersaline environments at different sea levels [28,29,30], such as in saline soils [31], marine solar saltern [32], and in deep subsurface or subseafloor hypersaline environments [33,34,35]. According to Macey et al. (2020) representatives of the *Marinobacter* genus are capable of incomplete sulfur oxidation as a source of energy [36].

Next, the ARDRA group IV comprised 18 strains, being the second most abundant bacterial population after *Guyparkeria* in this study. The sequenced strain showed the highest similarity to a typical halophilic genus *Halomonas*, belonging to the order *Oceanospirillales. Halomonas* are morphologically and metabolically diverse bacteria, commonly isolated from various hypersaline habitats, mainly saline lakes, salterns, saline soils, salt mines [37,38,39,40], deep subsurface sediments, or reservoirs [34,41]. Representatives of the *Halomonas* genus can oxidize reduced sulfur compounds and use thiosulfate as the supplemental energy source [42]. The phylogeny of the genus *Halomonas* is complex, with more than 120 validly published species (https://lpsn.dsmz.de/genus/halomonas, accessed on 1 May 2022). The 16S rRNA gene of our sequenced strain (3SP14B1) shared no more than 98.47% similarity to *H. ventosae* and 98.69% *to H. sediminicola* when compared against rRNA/ITS GenBank or EzTaxon databases, respectively. This is the only sequence comparison for which different affiliations were obtained. The difference could be explained by different sets of strains included in both databases and possibly due to different algorithms used for comparisons—local versus global alignment in rRNA/ITS and EzTaxon databases, respectively. Multiple sequence alignments later placed the 3SP14B1 strain clearly into the *H. sediminicola* cluster (see Figure 1). A low level of sequence similarity might suggest that this strain, and possibly all members of ARDRA group IV, belong to a potentially novel *Halomonas* species. Considering the fact that members of the genera *Halomonas* and *Marinobacter* are able to oxidize thiosulfate [35,42], the reduced sulfur compounds present in brine can serve as an alternative energy source to support the growth of these bacteria in such an oligotrophic environment.

The last identified bacterial genus of the cultivable, sulfur-oxidizing microbiota of Solivar salt brine was *Hydrogenovibrio*, formerly known as *Thiomicrospira* (order *Thiotrichales*). The representatives of this genus were clustered into two distinct groups based on their ARDRA profiles (V and VI), and follow-up analyses of the 16S rRNA genes revealed that the closest relative is *H. crunogenus*. However, low levels of sequence similarity to rRNA/ITS GenBank entries (97.55 and 98.19%) indicated again that these strains can be considered new *Hydrogenovibrio* species. *Hydrogenovibrio* are obligate chemolithoautotrophic, sulfur- and hydrogen-oxidizing bacteria, often isolated from deep-sea hydrothermal vents [43,44,45]. The presence of *Hydrogenovibrio* species in water from a sub-surface hypersaline environment in Solivar is consistent with previously published data showing that these bacteria are exceptionally well adapted for growth in harsh conditions, characterized by the scarcity of organic carbon sources and the presence of inorganic molecules as the only available electron donors [43,45,46].

To visualize evolutionary relationships, a phylogenetic tree based on 16S rRNA sequence data of isolated halophilic sulfur-oxidizing bacteria and their closest relatives (best BlastN search hits) was created (Figure 1).

In accordance with sequence analysis results, each 16S rRNA gene sequence was clustered with respective genes of other members of the genus. The length of branches along with the high bootstrap values of branching points support the assumption that at least some of the strains analyzed in this study represent novel bacterial species.

## 4. Conclusions

Deep-subsurface aquifers belong to one of the most unexplored habitats. Brine from the former salt mine Solivar can be considered as an extreme environment due to a range of challenging physical and chemical parameters, represented mainly by high salt concentration and oligotrophic conditions. In this study, cultivable sulfur oxidizers were obtained using thiosulfate agar medium in pure cultures, which were tested for their ability to tolerate different concentrations of salt as well as for their ability of autotrophic/heterotrophic growth. Six species of halophilic bacteria, members of *Marinobacter*, *Halomonas*, *Hydrogenovibrio*, and *Guyparkeria* genera, all with the ability to oxidize reduced sulfur compounds, were identified. Based on 16S rRNA sequence comparison, at least some of the strains might represent novel bacterial species. This preliminary research provides one of the first insights into the population structure of cultivable sulfur-oxidizing bacteria, which are still poorly understood and are an under-researched part of hypersaline environments. The data obtained also indicate that deep-subsurface hypersaline habitats harbor a largely unexplored microbiota with novel taxa of extremophilic bacteria. However, further metagenomics and culturomics studies are needed for a comprehensive assessment of the whole microbial biodiversity, including bacteria as well as archaea.

## Figures and Tables

**Figure 1 microorganisms-10-00995-f001:**
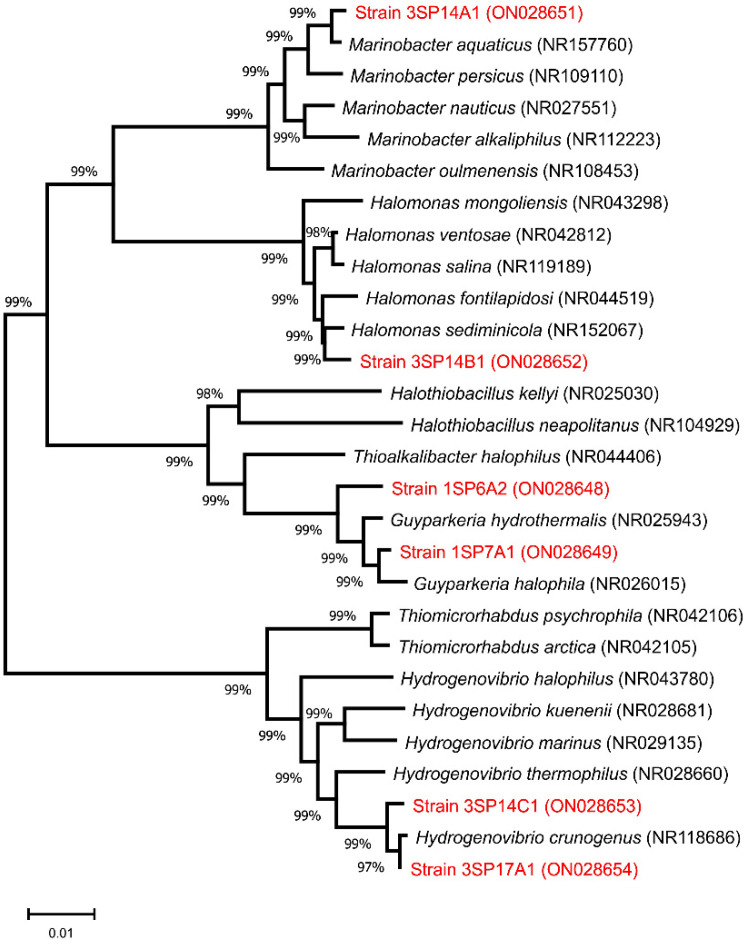
Unrooted neighbor-joining phylogenetic tree showing the relationship of 16S rRNA gene sequences of bacteria isolated in this study (red) with genes of their closest relatives. The scale represents evolutionary distances, which are in the units of the number of base substitutions per site.

**Table 1 microorganisms-10-00995-t001:** Identification of representatives of ARDRA (Amplified Ribosomal DNA Restriction Analysis) groups based on 16S rRNA gene sequencing and results of cultivation analysis using different concentrations of NaCl.

ARDRA Group	Number of Strains	Sequenced Strain	GenBank Accession Number	BlastN Best Hit	Similarity (%)	EzTaxon	Similarity (%)	Autotrophic Growth on Thiosulfate Medium with NaCl (%)		Heterotrophic Growth on R2A Medium with NaCl (%)		
								10	20	5	10	20
I	7	1SP6A2	ON028648	*Guyparkeria hydrothermalis*	97.63	*Guyparkeria hydrothermalis*	97.71	+	-	-	-	-
II	11	1SP7A1	ON028649	*Guyparkeria halophila*	99.04	*Guyparkeria halophila*	99.12	+	-	-	-	-
III	3	3SP14A1	ON028651	*Marinobacter aquaticus*	99.49	*Marinobacter aquaticus*	99.49	+	+	+	+	-
IV	11	3SP14B1	ON028652	*Halomonas ventosae*	98.47	*Halomonas sediminicola*	98.69	+	+	+	+	-
V	3	3SP14C1	ON028653	*Hydrogenovibrio crunogenus*	97.55	*Hydrogenovibrio crunogenus*	99.27	+	-	-	-	-
VI	6	3SP17A1	ON028654	*Hydrogenovibrio crunogenus*	98.19	*Hydrogenovibrio crunogenus*	99.93	+	-	-	-	-

## Data Availability

All 16S rRNA data obtained through this study were deposited into the GenBank database.

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
