# Peer review of "Deep Subsurface Hypersaline Environment as a Source of Novel Species of Halophilic Sulfur-Oxidizing Bacteria"

_microorganisms, 2022, doi:10.3390/microorganisms10050995_

Round 1

Reviewer 1 Report

The communication describes the isolation and identification of sulfur-oxidizing halophilic bacteria from deep subsurface hypersaline environment located in the former salt mine in Solivar (Presov, East Slovakia). The geological formation of the studied environment is well described. Different species of halophilic bacteria were identified by culture and molecular approaches, and also a phylogenetic analysis was conducted. This research provides an insight into the population of cultivable sulfur-oxidizing bacteria among halophilic microorganisms in an extreme habitat.

The manuscript is clearly presented and well-organized. The study is carried out in deep-subsurface aquifers in a salt mine, which is an interesting extreme environment. However, the results are not very relevant, since only a small number of halophilic bacteria are identified based on the 16S RNA coding gene, and most of them show a high percentage of similarity with the sequences from databases. A more complete study could have been achieved using metataxonomy or metagenomics approaches combined with culturomics to identify the whole microbial diversity. In addition, some special aspects of the isolates could have been described to give this work a novelty. Finally, the role of the sulfur-oxidizing bacteria and their value in the environment could be discussed.

The methods are clearly described, the references are appropriate to the content and some recent studies are cited in order to compare the obtained results. Nonetheless, the information reported in this manuscript is very limited. Only a few bacterial strains are isolated and identified. More species could be identified in order to determine the microbial diversity of this group of halophiles, including the use of different isolation culture media or microbial enrichment procedures for example.

With respect to the isolation method of halophilic sulfur-oxidizing bacteria, only one type of isolation medium is used and it is not clear how the sulfur-oxidizing activity is measured. There is no mention of pH changes as an indicator of the thiosulfate-oxidizing activity or any other method to reveal the reported sulfur-oxidizing activity.

The authors say that the salinity of the water sampled was around 30%, so, why did they only use a medium with 5% salinity for the isolation? To harbor, a greater diversity of halophiles higher concentrations of salts may be tested in the isolation medium, since extreme halophiles need higher salinity to proliferate. Could this issue be the reason why in the following experiment most of the isolated bacteria did not grow above 10% salinity? Maybe extreme halophilic bacteria and/or archaea have been discounted.

To study autotrophic/heterotrophic growth, diverse carbon sources could be added to the thiosulphate medium in order to see differences.

Concerning the molecular approach, the ARDRA analysis used for dereplication seems to not be very accurate, since two different ARDRA groups resulted in the same species. Were the banding patterns different? The banding patterns could be shown.

Chao1 richness is mentioned to justify that the whole diversity of cultivable halophilic sulfur oxidizers is covered, however, the data is not shown. How was it calculated? Which was the value of this biodiversity index?

Line  32 “Archaebacteria” is an obsolete name for Achaea.

Line 146 “obligately halophilic” seems to be redundant as halophilic already indicates the need for salt to grow.

I found DNA sequences ON028648 - ON028654 as not visible at GeneBank.

Fig 1. In the phylogenetic tree based on 16S rRNA, the outgroup needed to root the tree is missing.

In the legend of Fig 1, there is no indication of the bar with the value of 0.01. What does this bar mean?

To conclude that “cultivable sulfur oxidizers were obtained” more experiments are necessary to confirm that the isolates effectively oxidize the thiosulfate present in the medium to sulfate.

Reviewer 2 Report

The paper "Deep Subsurface Hypersaline Environment as a Source Novel Species of Halophilic Sulfur-Oxidizing Bacteria" is of high interest especially for microbiologist working in extrem environments.  The paper describes the isolation of 41 sulfur-oxidizing bacteria and identification of 6 isolates based on 16S sequencing. The paper appears to be one of the first studies describing culturable sulfur-oxidizing bacteria collected from the brine of a salt mine.

The paper is overall well written, however, some section need improvement and as such the paper would benefit  if it were to be edited by a native English speaker.

Details concerns and comments are below. 

Introduction:

The authors indicate that based on pilot studies diverse populations of halophilic bacteria were observed in the brine from the salt mine and mention that 80% of isolates were able to grow on thiosulfate medium.  It would be better to present the data in the paper.

Material and Methods  
The material and methods should be more detailed.  For example, include where chemicals were purchased, and exact composition of the medium

Line 47: How many water samples were collected, and at what approximate depth?

Line 56: the composition of the thiosulfate agar is incomplete, it is missing sodium thiosulfate. Furthermore, the environmental pH is listed at pH 6.5, and thesalinity (TDS) at 311g/L.  What is the rational to culture the bacteria at pH 7.5 and at a fairly low salinity (50g/L)? 

Line 57: At what temperature where the plates incubated? 

Line 59:  How often were the colonies sub-cultured?

Line 69: for the DNA extraction, what where the culturing conditions (i.e., medium, volume of culture, as well as RPM of incubator).

Line 82: How long was the initial denaturation?

Line 86 and following: 

How were the PCR amplicons purified for the restriction digest?  
For ARDRA, what was the rational for choosing AluI and HaeIII?

Also, if 2 digestions are prepared, did both show the same grouping or was it a double digestion?

How did you visualize the gels and how was the banding pattern analyzed?

Is it possible to submit a figure showing an example of the banding patterns?

Line 105: I am not sure that the data allows to estimate species richness.

Result and Discussion

The description of the Salt mine needs improvement.  It is not clear.

Line 127:  The sentence implies that there were different levels of salinties.  What was the range of salinities?   

Section 3.1, Cultivation analysis

You describe in detail growth condition (type of medium, NaCl concentrations) of all 41 isolates. Based on ARDRA you identified 6 groups and in table 1 show growth conditions of the 6 representative isolates that were further analyzed by 16S sequencing. Did all isolates of a specific ARDRA group have the same growth conditions?  And could you provide a table (maybe as a supplement) that shows the growth conditions of all isolate?

Table 1: While you present growth data for 5% NaCl for the R2A medium, 5% NaCl data for the thiosulfate agar is missing.  Please include the data.  Since you mention in the text that no isolate was growing at 0% NaCl, you could drop the column.  If you choose to keep the column, I would suggest to also include a 0% NaCl column for the R2A medium.

Line 170:  As mentioned above, I am not sure whether you can estimate reliably the species richness based on your study.  You isolated bacteria using a 5% salt concentration and at pH 7.5.  The pH of the initial water was 6.5, and TDS 311 g/L.  Thus it is conceivable that changing some of the conditions in your initial isolation step may reveal a few isolates that you have not captured. Thus I would recommend to remove that statement.

Line 191:  As you know, % similarity levels of 16S as an indicator for new species and genera is controversial.  Thus, to err on the safe side I would suggest that indeed 97% similarity suggests possibly a new species.  However, I would be hesitant to go as far as suggesting that your isolate may be a new genus just based on 16S.  Especially since later in the discussion (line 220) you suggest a new species for Hydrogenovibrio based on a 97% similarity, but do not suggest a new genus. 

Conclusion:

Line 252 and following: While I agree that it is quite possible that the deep surface habitat may reveal unexplored and unidentified bacteria and archaea, you focused your study on culturable sulfur oxidizing bacteria. Since you did not present data based on culture independent approaches (next generation sequencing 16S amplicon studies or metagenomic studies), I suggest to either remove the statement or rephrase the statement to indicate that culture independent studies could reveal a largely unexplored microbiom in this salt mine.

Reviewer 3 Report

The manuscript by Nosalova and co-workers deals with first stage of investigation of halophilic sulfur-oxidizing bacteria inhabiting poorly studied extreme environment – a salt mine in Solivar (Presov, East Slovakia). Using water from a borehole at a former salt mine authors succeeded in isolation of 41 bacterial strains growing by thiosulfate-oxidation both in autotrophic and heterotrophic conditions. Strains were preliminary divided by ARDRA analysis to six groups and then representative strain from each group was identified by 16S rRNA gene sequencing. Comparison of obtained 16S rRNA gene sequences with those in database allowed to include 26 autotrophic thiosulfate-oxidizers to the genera Guyparkeria and Hydrogenovibrio, and 15 heterotrophic strains – to genera Halomonas and Marinobacter. Growth of the strains at various salinity (0, 5, 10, 20% (w/v) NaCl) was determined and 16S rRNA gene sequences were deposited to the GenBank database. The work was carried out using microbiological and the simplest molecular methods, but the isolated strains of both autotrophic and heterotrophic bacteria, some of which belong to new taxa, are of undoubted value. Currently, most researchers are limited to molecular analysis of the composition of the microbial community, while fewer and fewer microbiologists are able to isolate unique bacteria.

The article by Nosalova and co-workers may be published in the Microorganisms journal after minor revision.

Comments of the Reviewer are given below and indicated in the marked manuscript.

Could the authors clarify in the article from what depth the examined water sample was taken?

Could the authors discuss in the article which possible substrates supported the growth of heterotrophic strains of the genera Halomonas and Marinobacter in their environment?

Since the authors claim in the article that they have isolated 41 pure cultures, it is better to call them “Strains” in the text of the article and in the figure, since they will probably be described as new taxa under the same numbers.

The quality of Figure 1 should be improved.

There are typos in the list of references.

Since there is a geological characteristic of the habitat in the articles in English, the reviewer is not sure how much value the references in Slovak are for readers.

Reviewer 4 Report

The MS submitted by Nosalova and coworkers explores the use of Deep Subsurface Hypersaline Environment as a Source of Novel Species of Halophilic Sulfur-Oxidizing Bacteria. The topic is of interest and fits the scope of the journal. The MS is well written and organized. Some minor issues related to grammar and spelling must be revised.

Round 2

Reviewer 1 Report

The communication “Deep Subsurface Hypersaline Environment as a Source of Novel Species of Halophilic Sulfur-Oxidizing Bacteria” has been improved according to the reviewer’s comments.

Thanks to the authors for the clarification of the items highlighted in the manuscript and their response. I understand that this study is a preliminary work about the potential of the salt mine in Solivar to harbor cultivable halophilic sulfur-oxidizing bacteria more than the characterization of the whole diversity of sulfur-oxidizing microorganisms.

I still believe that more microbial species including Archaea could have been isolated using different salt concentrations in the isolation medium. Maybe it could be covered in further studies.

With respect to the selection of sulfur-oxidizing microorganisms, I think that the information related to the production of sulfur nanoparticles during microbial growth is relevant and should be added to the text.
